# Consecutive-Day Ventricular and Atrial Cardiomyocyte Isolations from the Same Heart: Shifting the Cost–Benefit Balance of Cardiac Primary Cell Research

**DOI:** 10.3390/cells11020233

**Published:** 2022-01-11

**Authors:** Joachim Greiner, Teresa Schiatti, Wenzel Kaltenbacher, Marica Dente, Alina Semenjakin, Thomas Kok, Dominik J. Fiegle, Thomas Seidel, Ursula Ravens, Peter Kohl, Rémi Peyronnet, Eva A. Rog-Zielinska

**Affiliations:** 1Institute for Experimental Cardiovascular Medicine, University Heart Center Freiburg—Bad Krozingen and Faculty of Medicine, Albert-Ludwig University of Freiburg, 79110 Freiburg im Breisgau, Germany; joachim.greiner@uniklinik-freiburg.de (J.G.); teresa.schiatti@uniklinik-freiburg.de (T.S.); david.kaltenbacher@uniklinik-freiburg.de (W.K.); alina07@gmx.de (A.S.); thomas.kok@uniklinik-freiburg.de (T.K.); ursula.ravens@uniklinik-freiburg.de (U.R.); peter.kohl@uniklinik-freiburg.de (P.K.); 2Department of Experimental and Clinical Medicine, Division of Physiology, University of Florence, 50134 Florence, Italy; marica.dente@unifi.it; 3Institute of Cellular and Molecular Physiology, Friedrich-Alexander-University of Erlangen-Nürnberg, 91054 Erlangen, Germany; dominik.fiegle@fau.de (D.J.F.); thomas.seidel@fau.de (T.S.); 4CIBSS Centre for Integrative Biological Signalling Studies, University of Freiburg, 79110 Freiburg im Breisgau, Germany

**Keywords:** cardiomyocyte isolation, single cell, sarcomere shortening, tissue preservation

## Abstract

Freshly isolated primary cardiomyocytes (CM) are indispensable for cardiac research. Experimental CM research is generally incompatible with life of the donor animal, while human heart samples are usually small and scarce. CM isolation from animal hearts, traditionally performed by coronary artery perfusion of enzymes, liberates millions of cells from the heart. However, due to progressive cell remodeling following isolation, freshly isolated primary CM need to be used within 4–8 h post-isolation for most functional assays, meaning that the majority of cells is essentially wasted. In addition, coronary perfusion-based isolation cannot easily be applied to human tissue biopsies, and it does not straightforwardly allow for assessment of regional differences in CM function within the same heart. Here, we provide a method of multi-day CM isolation from one animal heart, yielding calcium-tolerant ventricular and atrial CM. This is based on cell isolation from cardiac tissue slices following repeated (usually overnight) storage of the tissue under conditions that prolong CM viability beyond the day of organ excision by two additional days. The maintenance of cells in their near-native microenvironment slows the otherwise rapid structural and functional decline seen in isolated CM during attempts for prolonged storage or culture. Multi-day slice-based CM isolation increases the amount of useful information gained per animal heart, improving reproducibility and reducing the number of experimental animals required in basic cardiac research. It also opens the doors to novel experimental designs, including exploring same-heart regional differences.

## 1. Introduction

Single cells, isolated from various tissues, have been a foundation for the development of mechanistic insight into many organ systems, promoting the understanding of physiology, pathology, and therapeutic interventions. Specifically, in experimental cardiac research, CM isolated from human and animal hearts have become indispensable over the 50 years since the first description of Langendorff perfusion-based enzymatic CM isolation [1].

In general, primary adult CM isolation is performed after sacrificing the experimental animal. While every successful isolation yields millions of CM, most of these cells can be used for functional research only over a relatively short time period. Because most commonly employed functional CM assays (such as electrophysiology, functional imaging, or mechanics measurements) are low-throughput, the vast majority of isolated viable CM are usually discarded. Therefore, for each set of single CM experiments, an animal is sacrificed, although each heart contains enough CM for tens, if not hundreds, of functional experiments.

Such wastefulness arises additionally from the fact that the quality of freshly isolated CM declines swiftly after liberation from tissue. Adverse structural and functional remodeling can be seen as early as 4–8 h post-isolation. The detrimental changes observed in CM within hours of isolation include CM swelling and deformation [2,3,4], changes in whole-cell capacitance [5], remodeling of trans-sarcolemmal and intracellular ion handling [6,7,8], depressed electro-mechanical function [9], internalization of surface proteins [10], loss of caveolae and transverse tubules [7], myofilament atrophy [3], or mitochondrial swelling and rupture [2,5]. The factors contributing to the steep decline in quality of freshly isolated CM are thought to include the elimination of mechanical support structures (cell–cell and cell–matrix coupling), the elimination of paracrine signaling, and the cessation of electrical stimulation and mechanically loaded contractions. This progressive cell remodeling curtails the relevance of culturing freshly isolated CM as a reproducible and representative experimental model system. In contrast, while immortalized or stem cell-derived cardiac cell lines are commercially available and sustainable in culture, they display an immature phenotype lacking homology to many structural and functional features of adult CM [11].

Here we develop and validate a method of extending both atrial and ventricular cell-in-tissue lifetime and quality by conducting multi-day CM isolations from the same rabbit heart. We show that live tissue storage [12] can be combined with slice-based CM isolation [13] to yield CM from the same donor heart over multiple days, circumventing the adverse remodeling in transcript levels, microscopic structure, and contraction and relaxation kinetics that are otherwise seen in isolated CM over the same period of time. Our method offers new opportunities for the dissection of regional differences in CM properties, and it may be applied to tissue that cannot be perfused (i.e., human or large animal heart biopsies). Multi-day CM isolation therefore offers a means towards extending the time period over which high-quality single-cell research can be conducted on adult primary CM, while at the same time lowering intra-individual variability and contributing to the aim of reducing the number of animals needed for experimental research.

## 2. Materials and Methods

### 2.1. Animals

All experiments were carried out according to the guidelines stated in Directive 2010/63/EU of the European Parliament on the protection of animals used for scientific purposes and were approved by the local authorities in Baden-Württemberg (X-16/10R). New Zealand white rabbits were used in this study (*n* = 18, age ≈ 10 weeks, both sexes, weight ≈ 1700 g). Rabbits were anesthetized via intramuscular injection of esketamine hydrochloride (Ketanest S 25 mg/mL, Pfizer Pharma PFE GmbH, Berlin, Germany; 12.5 mg/kg body weight) and xylazine hydrochloride (Rompun 2%, Bayer Vital GmbH, Leverkusen, Germany; 0.2 mL/kg body weight). During anesthesia, 1000 units of heparin (Heparin Sodium 5000 I.U./mL, B. Braun Melsungen AG, Melsungen, Germany) and 5 mg of esketamine hydrochloride were given intravenously. Thiopental (Thiopental Inresa 0.5 g, Inresa Arzneimittel GmbH, Freiburg, Germany; 12.5 mg/mL) was then injected intravenously until apnea.

### 2.2. Cardiac Slice Preparation

An overview of the protocol steps is provided in Figure 1. After excision, the heart was Langendorff perfused for ~1 min (until cessation of contractions) with warm (37 °C) modified Tyrode’s solution (flow rate 20 mL/min) containing (in mM): 138 NaCl, 0.33 NaH_2_PO_4_, 5.4 KCl, 2 MgCl_2_, 10 HEPES, 10 glucose, 0.5 CaCl_2_, and 30 2,3-butanedione monoxide; pH 7.3, 300 mOsm. This step can be omitted when processing tissue that cannot be perfused.

The heart was then dissected in cold (4 °C) modified Tyrode’s solution; the left ventricular free wall, left atrial free wall, and left atrial appendage were used for further processing. The tissue was cut into blocks with an X/Y size of up to 1 × 1 cm^2^ (ventricle)/0.5 × 0.5 cm^2^ (atria). These were embedded in 4% low melting point agarose (Carl Roth, Karlsruhe, Germany) prepared with modified Tyrode’s solution, cooled to 4 °C, and mounted on the stage of a vibratome (7000-smz-2 with 7550-1-SS blade, both Campden Instruments Ltd., Loughborough, UK) using cyanoacrylate-based low-viscosity glue. During mounting, care was taken to visually estimate the prevailing orientation of CM within the slice, and to mount the tissue so that the cutting plane would be parallel to the epicardial surface of the tissue block. The tissue was submerged in ice-cold modified Tyrode’s solution bubbled with O_2_, and sliced into 300 µm thick sections at a vertical blade vibration frequency of 60 Hz (ventricle) and 80 Hz (atria), an amplitude of 1.5 mm, and an advancement velocity of 40 µm/s (ventricle) and 10 µm/s (atria). Ventricular tissue was freshly sliced from tissue chunks on each day before CM isolation, whereas atrial tissue was sliced on the day of heart excision and the slices were stored until CM isolation. All tissue was stored in modified Tyrode’s solution at 4 °C and under room atmosphere, for up to three days.

### 2.3. Cell Isolation

CM isolation from slices was adapted based on a previously published protocol [13]. In brief, the agarose around the slice was gently removed, and the slices were placed in 35 mm diameter plastic Petri dishes and washed in modified Tyrode’s solution. Slices were brought up to room temperature, then rinsed with wash solution 1 containing (in mM): 100 NaCl, 2.5 KH_2_PO_4_, 15 KCl, 2 MgCl_2_, 10 HEPES, 10 glucose, 20 taurine, 20 L-glutamic acid monopotassium salt, and 30 2,3-butanedione monoxide (BDM), supplemented with 2 mg/mL bovine serum albumin; pH 7.3, 300 mOsm. Dishes were then placed on a custom-built heated orbital shaker. All isolation steps were performed with constant rocking at 65 rpm, and at 37 °C. Tissue was digested for 12 min using proteinase mix XXIV (concentration: 0.5 mg/mL in 2 mL wash solution 1; Sigma-Aldrich, St. Louis, MO, USA). Tissue was then rinsed twice in wash solution 1 (each 2 mL), and further digested with Liberase TL Research Grade (concentration: 0.23 mg/mL in 2.2 mL wash solution 1 supplemented with 5 µM CaCl_2_; Hoffmann-La Roche, Basel, Switzerland) for up to 45 min (ventricles)/15 min (atria). Tissue was then transferred into 2 mL of wash solution 2 (the same as wash solution 1, but with bovine serum albumin content increased to 10 mg/mL, and supplemented with 5 µM CaCl_2_) and gently dissociated mechanically, using forceps to free individual CM. Calcium recovery and partial BDM washout were done by a stepwise (usually in 9 steps) addition of modified recording Tyrode’s solution containing (in mM): 137 NaCl, 4 KCl, 1 MgCl_2_, 10 HEPES, 10 glucose, 1.8 CaCl_2_, 20 taurine, 10 creatine, 5 adenosine, and 2 L-carnitine; pH 7.3, 300 mOsm. This was continued until a final concentration of 1 mM Ca^2+^ and 12.8 mM BDM were reached. The time interval between steps was 2 min (ventricles) and 5 min (atria).

The cell suspension was then filtered through mesh (filter size: 1 mm × 1 mm) and CM were allowed to settle for 15 min in a 15 mL centrifuge tube. The supernatant was removed, and the CM were re-suspended in modified recording Tyrode’s solution (removing residual BDM and bringing the Ca^2+^ concentration to 1.8 mM). Composition of all solutions used for CM isolation can be found in Appendix A.

### 2.4. Assessment of Cell Isolation Yields

Live CM were imaged post-isolation using a Leitz Laborlux 11 microscope (Leica Microsystems, Vienna, Austria) with a 2.5× air objective and an eyepiece camera (MikrOkular, Bresser, Rhede, Germany). To assess the yield of rod-shaped CM, we counted rod-shaped and dead (rounded) CM in at least three fields of view per isolation (60–500 CM manually counted per field of view). To assess contractile function, we paced CM in a perfusion chamber equipped with field stimulation electrodes (RC-27N, Warner Instruments, Hamden, MA, USA), driven by a MyoPacer field stimulator (applied voltage was twice the threshold and maximally 12 V, pulse duration 5 ms; pacing frequency was 0.5 Hz for ventricles and 1 Hz for atria; IonOptix, Westwood, MA, USA). Fractions of rod-shaped and contracting CM (i.e., responding to field stimulation without arrhythmic events, e.g., pauses, extra contractions) are expressed as percent of all CM in a field of view.

### 2.5. High-Resolution Imaging

To assess the regularity of cross-striations and the potential presence of membrane ‘blebbing’ (evidence of local detachment of the membrane from the cytoskeleton), high-resolution imaging was performed using a Leica DMi8 microscope with a 60× oil objective for the ventricular CM, and with a Leica TCS SP8 X laser scanning confocal microscope (both Leica Microsystems, Vienna, Austria) using a 40× water immersion objective.

### 2.6. Sarcomere Shortening Dynamics

Unloaded CM shortening was assessed optically, as described previously [14]. Briefly, contractions were triggered by field stimulation, as described above, at room temperature. CM were imaged using an inverted microscope with a 40× air objective using phase contrast (Leica DM IRBE microscope; Leica Microsystems, Vienna, Austria). Sarcomere shortening dynamics were quantified using the IonWizard 6.6 software package (IonOptix, Westwood, MA, USA). CM with a resting sarcomere length (SL) shorter than 1.65 µm were excluded from the analyses.

### 2.7. Resting Membrane Potential Recordings

The patch-clamp technique was used to investigate the membrane potential of isolated CM. Experiments were performed at room temperature with a patch-clamp amplifier (200B, Axon Instruments, Foster City, CA, USA) and a Digidata 1440A interface (Axon Instruments). Signals were digitized at 10 kHz, low-pass filtered at 1 kHz, and analyzed with pCLAMP 10.3 (Axon Instruments) and OriginPro 2021b (OriginLab, Northampton, MA, USA) softwares. Whole-cell recordings in current-clamp mode were obtained with the following bath solution (in mM): 140 NaCl, 5.4 KCl, 1 CaCl_2_, 2 MgCl_2_, 10 glucose, 10 HEPES; pH 7.4, 300 mOsm. The pipette solution contained (in mM): 50 KCl, 80 K-aspartate, 2 MgCl_2_, 3 Mg-ATP, 10 EGTA, 10 HEPES; pH 7.4, 300 mOsm, as used in previous work [15]. Membrane potential values are given after correction for liquid junction potential. All membrane potentials were recorded at steady state, within 3–5 min after obtaining the seal. In all cases, immediately after obtaining the whole-cell configuration the membrane potential was negative and stable.

### 2.8. RNA Isolation, Reverse Transcription, and Quantitative PCR

RNA isolation was performed using the RNeasy Mini Kit (Qiagen, Venlo, The Netherlands). CM pellets were lysed immediately following isolation in buffer RLT (provided with the kit) supplemented with 1% β-mercaptoethanol. The lysate was processed through QIAshredder columns, and RNA isolated as per the manufacturer’s instructions. RNA was eluted in water, and quantity and quality were assessed spectrometrically. cDNA conversion was carried out using TaqMan Reverse Transcription Reagents (ThermoFisher Scientific Inc., Waltham, MA, USA) as per the manufacturer’s instructions, using 200 ng of RNA per reaction. Quantitative PCR (qPCR) was conducted using TaqMan pre-designed gene expression assays, Fast Advanced Master Mix (ThermoFisher Scientific Inc., Waltham, MA, USA), and a LightCycler 480 thermocycler (Roche, Pleasanton, CA, USA). Expression levels of voltage-dependent L-type calcium channel (LTCC) α2/δ subunit 1 (CACNA2D1, *Cacna2d1*, assay ID Oc03397798_m1), Na^+^/Ca^2+^ exchanger (NCX1, *Slc8a1*, assay ID Oc04250277_m1), and myosin heavy chain-β (MHC-β, *Myh7*, assay ID Oc03396451_m1) were assessed, and are expressed relative to the levels of mRNA encoding β-actin (*Actb* assay ID Oc03824857_g1).

### 2.9. Data Analysis

Data obtained from the same heart are indicated by color and presented as either individual points or as violin superplots including information on data heterogeneity—the normalized density estimates of individual replicates are stacked to show how each replicate (color-coded area) contributes to the overall density estimate (outline) [16]; all graphs additionally indicate the mean ± standard error of the mean (SEM). Statistical significance of data was assessed based on mean values of all daily recordings to avoid pseudo-replication. Data were analyzed using a paired one-way ANOVA with Tukey’s *post-hoc* test (unless stated otherwise). *p* < 0.05 was taken to indicate a statistically significant difference between means. 

## 3. Results

### 3.1. Yield of Calcium-Tolerant CM over Three Consecutive Days 

Comparisons of the yields of rod-shaped and of regularly contracting (in response to field stimulation) ventricular CM revealed no statistically significant differences between days 0 (heart excision), 1, and 2. The fractions of rod-shaped CM and of CM responding with visible contractions to electrical field stimulation were ≈60% and ≈50%, respectively, on all three days (Figure 2A,D). Similarly, the yields of rod-shaped and contracting atrial CM, both from the free wall and from the appendage, did not show statistically significant differences between isolation days, with a yield of ≈50% rod-shaped and of ≈40% contracting CM on all three days (Figure 2B,C,E). There were no statistically significant differences between yields obtained from atrial free wall and appendage.

### 3.2. Preservation of CM Morphology over Three Consecutive Days

Diastolic SL in CM contracting upon stimulation did not show statistically significant differences between isolations on consecutive days, with the mean SL > 1.7 µm both in ventricular and in atrial free wall and appendage CM (Figure 3A,C). Qualitative visual examination of both ventricular and atrial CM revealed regular cross-striations and a lack of membrane ‘blebbing’ in all rod-shaped cells, with no apparent morphological differences between CM isolated on three consecutive days from the same heart (Figure 3B,D and Appendix A).

### 3.3. CM Contraction and Relaxation over Three Consecutive Days

No statistically significant differences were observed in the degree of sarcomere shortening (Figure 4A,B), or maximum velocity of contraction or relaxation (Figure 4C) of ventricular CM isolated over three consecutive days from tissue slices from the same heart.

Similarly, for atrial CM (both from free wall and appendage), no statistically significant differences were observed with respect to the degree of SL shortening (Figure 5A,B), or maximum velocity of relaxation (Figure 5C) over three consecutive days of isolation. However, the maximum velocity of contraction on day 2 was lower than on day 0 (organ excision; Figure 5C). No statistically significant differences were found in the contractile parameters between the different regions of the atria.

### 3.4. Resting Membrane Potential over Three Consecutive Days

Resting membrane potential was measured using the patch-clamp technique and was found to not differ significantly between ventricular or atrial CM obtained over three consecutive days of isolation from the same heart (Figure 6).

### 3.5. Gene Expression Analysis over Three Consecutive Days

Transcript level analysis of genes involved in transmembrane calcium movement (encoding L-type Ca^2+^ channel α2/δ subunit 1 and Na^+^/Ca^2+^ exchanger), as well as in sarcomere integrity (myosin heavy chain-β) of ventricular CM revealed no statistically significant differences between three consecutive days of isolation (Figure 7). These genes were chosen as representative of both functional and structural characteristics of the CM.

## 4. Discussion

Here we present a method of extending the use of cardiac tissue for the isolation of primary ventricular and atrial CM past the day of organ excision. We demonstrate preservation of structural and functional properties of CM over three consecutive days of isolation from the same heart, thus allowing for reduction in experimental animal numbers in basic cardiac research.

We focus our method on rabbit CM, a model whose popularity in basic cardiac research is rapidly rising due to the similarities between rabbit and human cardiac architecture and electrophysiology [17]. To date, however, relatively few rabbit-specific experimental protocols have been made available. Using the method described here, functional CM can be isolated over multiple days following organ excision (currently three), allowing for multiple-day experiments using the same heart. The key advantage of using live tissue as a ‘cell micro-environmental incubator’ is the maintenance of a close-to-physiological structural micro-environment, including support of the mechanical load. In addition, the use of mechanical uncouplers helps to prevent tissue hypercontracture and preserve the ATP pool during cold storage [18], a concept similar to ‘cold cardioplegia’ used in transplantology [19,20,21].

In our study, we used BDM to mechanically uncouple the CM. Our choice was based on previous studies demonstrating positive effects of BDM on the viability of live cardiac tissue slices and isolated CM [22,23,24,25]. However, while BDM has been shown to improve preservation of CM contractility in comparison to blebbistatin, blebbistatin was shown to better preserve CM viability (defined as maintenance of rod-shaped morphology) [22]. In addition, BDM was suggested to have negative effects on mitochondrial respiration [26]. Therefore, the choice of the most suitable approach for future applications will depend on the individual research questions. As an additional alternative, N-benzyl-p-toluene sulphonamide (BTS) has been shown previously to extend the viability of single isolated canine CM [27].

In terms of CM quality assessment, the work presented here focuses on gross morphology, diastolic SL, contraction/relaxation kinetics, resting membrane potential, and transcript levels of several selected genes. A future in-depth analysis of CM mechanics (e.g., force generation), calcium handling mechanisms, CM metabolic profile and morphology (e.g., of transverse tubules or sarcomeres), as well as of large-scale gene expression and proteome profiles will be important to further understand the degree of CM preservation. In addition, prolonging the protocol to include further consecutive days is a goal, which may further increase the relevance of the method described here.

Crucially, the majority of rod-shaped CM—about 50% of ventricular and 40% of atrial CM—responded to field stimulation. Multi-day isolation of ventricular CM yielded structurally and functionally consistent cell populations. In the case of atrial CM, a reduced contraction velocity was observed on day 2. Atrial CM, however, exhibited no deterioration in any of the other parameters assessed. The reasons for a decline in atrial CM contraction velocity are currently unclear. It is possible that slight differences in storage (slices vs. chunks), isolation (duration of digestion) or recording protocols underlie these observations. Another possibility is that atrial CM are more susceptible to damage [28]. Further investigation of electrophysiological properties and an assessment of calcium handling mechanisms in atrial CM would help identify the most likely cause for the decline in contraction velocity seen on day 2 post-isolation. Refinement of the described protocol (also with other species in mind) may include the composition of solutions and specific CM isolation workflows for different heart regions. One possibility is the employment of the University of Wisconsin solution as a storage buffer—it was previously reported to successfully preserve both ventricular and atrial human tissue, and is thought to offer more precise control of osmotic and oncotic pressure, redox homeostasis, and metabolic turnover [29,30]. Alternatively, low sodium/high potassium solutions could be employed to prevent tissue hypercontracture during prolonged storage. In addition, a host of supplements have been previously suggested to enhance the survival of cultured CM—thyroid hormones, insulin, L-glutamate, creatine, L-carnitine, and taurine [21,31]. Finally, the viability of slices could be extended by culturing under constant electro-mechanical stimulation (shown to preserve slice function) between isolations [12,32,33,34].

Although slice-based isolation methods do not reach the yield obtained with coronary artery perfusion-based isolation methods [13], they provide multiple advantages. With a slice-based method, a clear separation between regions of interest is easily possible. Here, we separated the ventricles, atrial free wall, and appendage. Other conceivable applications include the investigation of remodeled vs. non-remodeled myocardium post-injury or sub-epicardial vs. sub-endocardial CM. Furthermore, slice-based methods can be adapted for multiple species, including humans, where coronary perfusion with enzyme-containing solution is usually not possible [13]. Finally, the adaptation of the slice-based isolation protocol to new species and tissues is easier compared to Langendorff based methods, allowing for multiple parallel and cost-saving comparisons of different experimental conditions (e.g., choice of enzyme, solution composition) using multiple fragments or slices from the same heart. Slice-based isolation methods would, however, be of limited use when larger numbers of CM are required (e.g., for membrane fractionation experiments or high-throughput drug testing), and they are also in general more time-consuming compared to Langendorff based isolation.

The major advantage of the presented method is its simplicity and ease of implementation in any laboratory. We believe that further development of the method will help researchers arrive at a ‘one-heart-per-working-week’ paradigm, instead of the current one-heart-per-experimental-day approach. This will save costs and lead to a significant reduction in the number of experimental animals, supporting the guiding principle of 3R (replacement, reduction, refinement).

## Figures and Tables

**Figure 1 cells-11-00233-f001:**
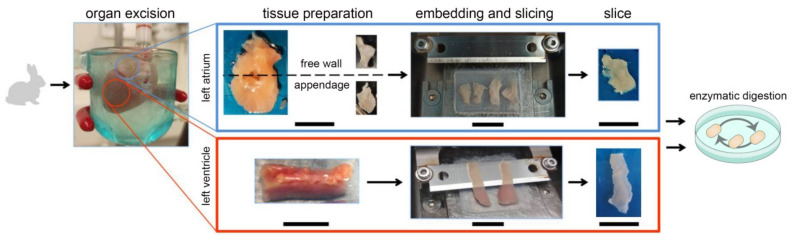
Overview of the protocol for the preparation of tissue slices and CM isolation from rabbit left atrium (**top**) and left ventricle (**bottom**), scale bars 10 mm.

**Figure 2 cells-11-00233-f002:**
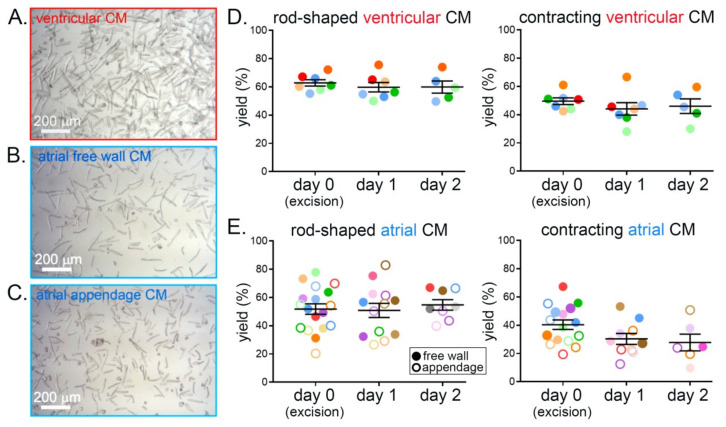
Yield of calcium-tolerant rabbit ventricular and atrial CM does not show statistically significant differences on three consecutive days of cell isolation from the same heart. (**A**) Representative brightfield microscopy image of ventricular CM after isolation. (**B**) Representative brightfield microscopy image of atrial free wall CM after isolation. (**C**) Representative brightfield microscopy image of atrial appendage CM after isolation. (**D**) Yield (percentage of all cells in the field of view) of rod-shaped and of contracting (upon stimulation; stimulation rate 0.5 Hz) ventricular CM on three consecutive days of isolation from tissue slices of the same heart; *n* = 5–7 hearts, 13,000–30,000 CM/day of isolation. (**E**) Yield of rod-shaped and of contracting atrial CM (stimulation at 1 Hz; free wall—filled circles; appendage—open circles); *n* = 3–10 hearts, 800–5700 CM/day of isolation. Each color represents data from one heart.

**Figure 3 cells-11-00233-f003:**
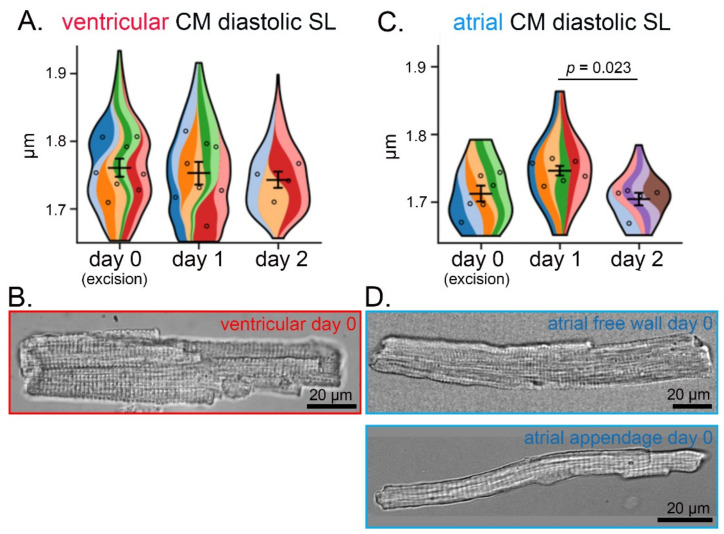
Ventricular and atrial CM morphology does not show statistically significant differences over three consecutive days of cell isolation from cardiac tissue slices from the same rabbit heart. (**A**) Diastolic SL of isolated ventricular CM over three consecutive days of isolation; *n* = 4–8 hearts, 150–300 CM/day of isolation. (**B**) Representative phase-contrast microscopy image of living ventricular CM after isolation. (**C**) Diastolic SL of isolated atrial (free wall and appendage combined) CM over three consecutive days of isolation; *n* = 5–6 hearts, 40–140 CM/day of isolation. (**D**) Representative brightfield microscopy image of living atrial free wall (**top**) and appendage (**bottom**) CM after isolation. (**A**,**C**) each color represents data from one heart. Within each colored area, the open circles represent the average of all cells recorded from the same heart.

**Figure 4 cells-11-00233-f004:**
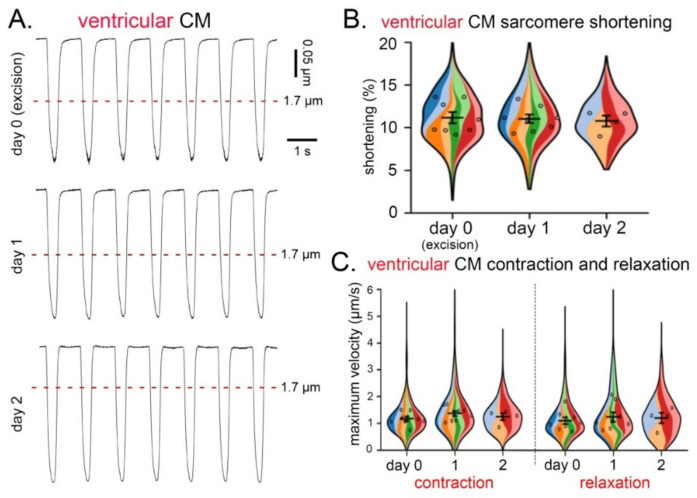
Contraction and relaxation of ventricular CM over three consecutive days of cell isolation from live cardiac tissue slices of the same heart. (**A**) Recordings of sarcomere shortening (in response to field stimulation at 0.5 Hz) over three consecutive days of CM isolation. (**B**) Degree of sarcomere shortening (%) in ventricular CM over three consecutive days. (**C**) Maximum velocity of contraction and relaxation in ventricular CM over three consecutive days of CM isolation; *n* = 4–8 hearts, 150–300 CM/day of isolation. (**B**,**C**), each color represents data from one heart. Within each colored area, the open circles represent the average of all cells recorded from the same heart.

**Figure 5 cells-11-00233-f005:**
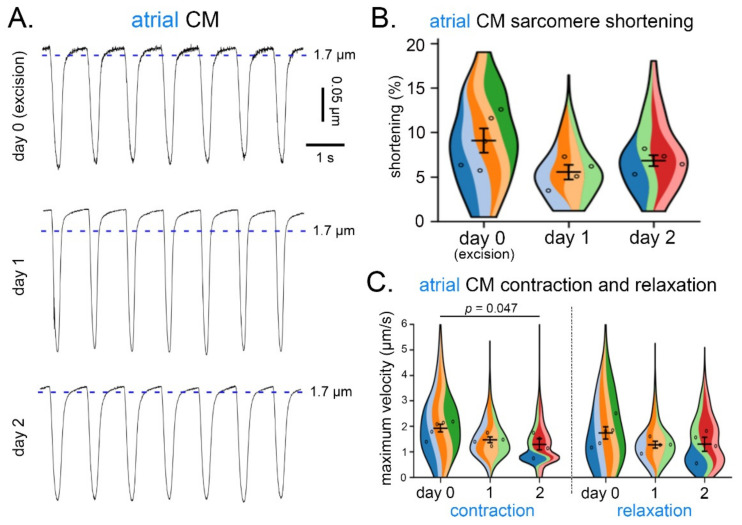
Contraction and relaxation of atrial CM over three consecutive days of cell isolation from live cardiac tissue slices of the same heart. (**A**) Recordings of sarcomere shortening (in response to field stimulation at 1 Hz) over three consecutive days of CM isolation from the same heart. (**B**) Degree of sarcomere shortening (%) in atrial CM (free wall and appendage combined) over three days of CM isolation. (**C**) Maximum velocity of contraction and relaxation in atrial CM (free wall and appendage combined) over three consecutive days of CM isolation; *n* = 4–5 hearts, 40–70 CM/day of isolation. (**B**,**C**), each color represents data from one heart. Within each colored area, the open circles represent the average of all cells recorded from the same heart.

**Figure 6 cells-11-00233-f006:**
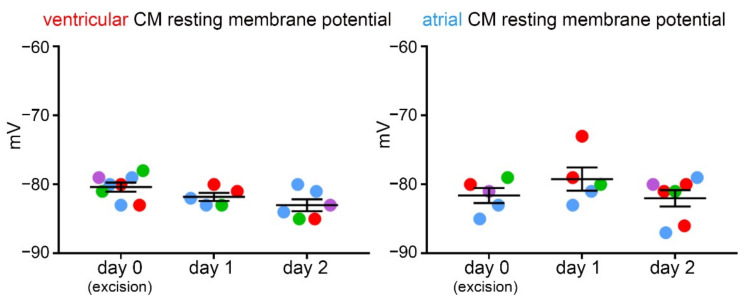
Resting membrane potential in ventricular and atrial CM (free wall and appendage combined) over three consecutive days of cell isolation from live cardiac tissue slices of the same heart; *n* = 5–8 CM/day of isolation from 3–4 hearts. Each color represents data from one heart.

**Figure 7 cells-11-00233-f007:**
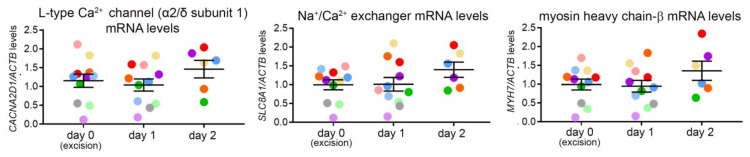
Ventricular CM gene expression over three consecutive days of cell isolation from live cardiac tissue slices of the same heart. Levels of mRNA, encoding L-type Ca^2+^ channel α2/δ subunit 1, Na^+^/Ca^2+^ exchanger, and myosin heavy chain-β, over three consecutive days of ventricular CM isolation. mRNA levels of the genes of interest are expressed relative to the level mRNA encoding β-actin (control transcript); *n* = 6–11 hearts/day of isolation. Each colored dot represents the average of data from the same heart.

## Data Availability

Data is available from the corresponding author upon request.

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
