# Peer review of "Consecutive-Day Ventricular and Atrial Cardiomyocyte Isolations from the Same Heart: Shifting the Cost–Benefit Balance of Cardiac Primary Cell Research"

_cells, 2022, doi:10.3390/cells11020233_

Round 1

Reviewer 1 Report

This manuscript describes a method of consecutive-day ventricular and atrial cardiomyocyte isolations from the same heart. The results show the morphology, contraction and relaxation, and gene expression of cardiomyocytes from three consecutive days.

The authors concluded that multi-day slice-based CM isolation increases the amount of helpful information gained per animal heart and also opens the doors to exploring the same heart regional differences. However, I have to say that the authors seem to over-claim their observations based on these presented data.

The high-quality isolation of adult cardiomyocytes is not an easy thing.  It is not easy to implement the technique in a new laboratory.

I have some concerns here which should be addressed by the authors:

The membrane potential of freshly isolated cardiomyocytes is lost due to undergoing the process of isolation. Thus, the membrane potential of fresh myocytes could serve as a quality indicator of the isolation. When is it properly to analyze the myocytes if adverse structural and functional remodeling happened as early as 4–8 h post-isolation? Comparing the membrane potential of consecutive-day myocytes is a valuable indicator.

Myocytes are high metabolic cells, and metabolism and calcium signals are very important for their functions. Do the metabolism and calcium signals change during consecutive-day isolation?

It is well known that the function of cultured adult cardiomyocytes will decrease quickly in several days. The authors use a method of multi-day (days 0 (heart excision), 1, and 2)CM isolation from the same animal heart, yielding calcium-tolerant ventricular and atrial. Whether authors could isolate calcium-tolerant cardiomyocytes at day3、day4(heart excision)?

Fig. 2A,C, Diastolic sarcomere length (SL) in contracting CM with mean SL > 1.7 μm both in ventricular and in 209 atrial free wall and appendage CM . It is because CM with resting SL shorter than 1.65 μm has been excluded from analyses. Why CM with resting SL shorter than 1.65 μm need to be excluded?

Author use BDM, not blebbistatin. Andrew R. Hall reported that cardiomyocytes culturing with BDM alone significantly reduced cell viability compared to control at 18h (BDM – 99.7 ± 0.2% cell death vs. Control 63.7 ± 6.6% cell death). In contrast, when cells were cultured with blebbistatin alone, cells survived for longer. Charles S Chung showed that Cells stored in BDM maintained morphology and contractile function for 48 h. Storage in blebbistatin maintained cell morphology for 72 h but inhibited contractility after 24h. Therefore, when the authors examined the function of the cardiomyocytes? Whether it is better to use BDM before 6h and blebbistatin between 6-18h?

As a method, please provide more details ( temperature, volume and time…) for others to follow up.

Line 221, Please reformat Figures D and E as A,B,C.

The authors should present the images of myocytes from day 0 to day 2. Are these myocytes from the same heart area? Does it make sense to compare the myocytes of different heart slices or regions?

Author Response

Please see our point-by-point response attached.

Reviewer 2 Report

The manuscript by Greiner et al. presents with a very appealing experimental approach to isolate cardiomyocytes from the same heart on consecutive days, which may pave the way to reduce the number of laboratory animals and enhance reproducibility in cardiovascular research. 

Specific comments:

_Figure 1: It would be important to have higher magnification insets from B, D and E panels, to enable better visualization of cell morphology.

_Figure 1C: although no statistical difference was found, it seems that atrial cardiomyocytes (CM) present a higher difference in the yield of contracting cells from day 0 to days 1-2, than ventricular CM. Could this be related with the data from Figure 4 C, in which the authors show a significant decrease in maximum velocity of contraction? Are the atrial CM more sensitive to storage than ventricular CM?

_Figure 2 and Figure 4: in graphs depicting data from atrial CM, only open circles are visible. Do these graphs refer only to appendage atrial CM, or does it include also free wall atrial CM? Please revise figure legends.

_Figure 2: authors should also include a representative image of atrial appendage CM.

_For cardiovascular research, many studies require cultivation of atrial/ventricular CM. Would CM isolated from day 1 or 2 be suitable for further cultivation? Would the isolation day impact on further culturing time?

_Did the authors try to apply this procedure to hearts from other species, namely, to isolate mice, rats, or human CM? Could the authors elaborate/discuss on possible modifications to this protocol to be able to apply to other species (enzymes, storage solutions...)?

Author Response

We thank the reviewers and the editor for their appreciation of our work, their thorough assessment of our manuscript, and the encouraging comments on how to improve the manuscript. As suggested by reviewers, we have edited and moved the original Figure S1 to the main body of the manuscript (new Figure 1), provided additional representative image of atrial appendage CM (now Figure 3), increased the number of observations for resting sarcomere length and contraction/relaxation kinetics for both atria and ventricles (now Figures 3-5), and included new data regarding resting membrane potential (new Figure 6). We also included additional experimental information in Materials and Methods and provide a more in-depth discussion of our results as requested.

Our responses to the reviewers' comments are indicated in the attached document in blue font. All changes in the manuscript are also indicated in blue.

Reviewer 3 Report

In this manuscript, the authors present a method for isolating functional primary ventricular and atrial cardiomyocytes from the same rabbit heart in consecutive days. This method helps reduce the amount of wasted cells while gaining more information from the same animal heart. Validation and implementation of this method by other researchers have the potential to improve efficiencies and reduce the number of experimental animals needed, which can be highly beneficial. The manuscript is well-written with detailed methods and clearly presented figures. For Figure S1: since the focus of this manuscript is on the method of isolating ventricular and atrial CMs on consecutive days, it is important to show the overview of the protocol in the main figures to help readers visualize the experiment. Consider moving Figure S1 into the main figure and expand it to include single cell isolation/digestion step after the tissue slices.  

Author Response

We thank the reviewer for these comments. We have now expanded and moved Figure S1 to the main body of the manuscript (new Figure 1).

Reviewer 4 Report

In this manuscript, the authors  provide a method of multi-day cardiomyocyte isolation from one and the same animal heart, yielding calcium-tolerant ventricular and atrial cardiomyocytes. The method is based on cell isolation from cardiac tissue slices following repeated (usually overnight) storage of tissue under conditions that prolong cardiomyocyte viability beyond the day of organ excision by two additional days. Structural and functional characteristics are described in detail, validating the approach.

Minor remark:

  1. There is no need to abbreviate cardiomyocyte or cardiomyocytes.
  2. Same remark for sarcomere length abbreviation.
  3. Same remark for the sodium/calcium exchanger. In general, take care that all non-standard abbreviations are explained at first use and that use of abbreviations is restricted to enhance readbility.

Author Response

We thank the reviewer for this positive assessment.

Minor remark:

  1. There is no need to abbreviate cardiomyocyte or cardiomyocytes.
  2. Same remark for sarcomere length abbreviation.

We use abbreviations mostly as means of efficient use of space within the figures. To be consistent we then also abbreviate CM and SL it in the text. We would prefer to keep this convention of abbreviating frequently used words, but will of course be happy to accept any editorial policy-based decision by the journal, should our paper be accepted.

Same remark for the sodium/calcium exchanger. In general, take care that all non-standard abbreviations are explained at first use and that use of abbreviations is restricted to enhance readbility.

We agree and no longer abbreviate NCX (or LTCC and MHC-β).

Round 2

Reviewer 1 Report

The authors have addressed most of my concerns, but I still have several concerns.

The membrane potential of freshly isolated cardiomyocytes is lost due to undergoing the process of isolation but unchanged over the 3 days.

Does the membrane potential go all the way down to a stable value?

Calcium signals change is essential for cardiac cells function; only contraction and relaxation-related data are not solid.   

Reviewer 2 Report

The manuscript was significantly improved following the reviewers suggestions. My only comment is that the functional assays should be complemented with phase contrast images of cardiomyocytes (atrial and ventricular) isolated not only at day 0, but also at day 1 and 2. This is key to ascertain whether any degree of remodeling/architectural change is present following this procedure.
